# CONCUR: BENCHMARKING LLMs FOR CONCURRENT CODE GENERATION

## ABSTRACT

Leveraging Large Language Models (LLMs) for code generation has increasingly emerged as a common practice in the domain of software engineering. Relevant benchmarks have been established to evaluate the code generation capabilities of LLMs. However, existing benchmarks focus primarily on sequential code, lacking the ability to effectively evaluate LLMs on concurrent code generation. Compared to sequential code, concurrent code exhibits greater complexity and possesses unique types of bugs, such as deadlocks and race conditions, that do not occur in sequential code. Therefore, a benchmark for evaluating sequential code generation cannot be useful for evaluating concurrent code generation with LLMs. To address this gap, we designed a benchmark CONCUR specifically aimed at evaluating the capability of LLMs to generate concurrent code. CONCUR consists of 115 concurrency problems and leverages formal methods techniques, namely model checking, to assess the correctness of the generated code. We conducted an evaluation of a range of LLMs on CONCUR, highlighting limitations of current models. Overall, our work provides a novel direction for evaluating the capability of LLMs to generate code with focus on concurrency.

## 1 INTRODUCTION

Large Language Models (LLMs) have demonstrated significant success across software engineering tasks, with code generation being one of their most common applications. This growing influence has spurred the creation of benchmarks to systematically evaluate model performance on program synthesis. Although existing benchmarks (Chen et al., 2021; Athiwaratkun et al., 2022; Yu et al., 2024) evaluate LLMs' code generation capabilities and propose corresponding metrics and datasets, these benchmarks are primarily limited to sequential code.

Concurrency plays a crucial role in the software industry (Hwu et al., 2008). The development of concurrent programs is known to present numerous challenges (Bianchi et al., 2017). Unlike sequential code, concurrent programming is more complex due to the interleaved execution of tasks or threads (Yu & Narayanasamy, 2009). Concurrent code introduces a host of unique challenges, including nondeterministic thread scheduling, synchronization requirements, and subtle correctness issues such as race conditions, deadlocks, and starvation. These concurrency-specific bugs are notoriously difficult to anticipate, diagnose, and resolve, even for experienced developers. As a result, evaluating LLMs only on sequential code leaves a critical blind spot in understanding their true capabilities and limitations. This prevents many code generation benchmarks from reliably assessing the capability of LLMs to generate concurrent programs. Static similarity metrics, such as CodeBLEU (Ren et al., 2020), capture surface-level overlap and often overlook semantic correctness, while unit-test-based dynamic evaluation cannot systematically explore nondeterministic thread schedules. Therefore, these approaches may incorrectly classify flawed concurrent programs as correct.

To address this gap, we propose CONCUR, the first benchmark dedicated to evaluating LLMs on concurrent code generation. CONCUR consists of 43 carefully curated concurrency problems paired with structured prompts and validated ground-truth implementations in Java. We have also added 72 validated mutant problems from 24 (out of the original 43) eligible problems, resulting in a total of 115 problems. Beyond dataset design, CONCUR leverages formal methods techniques, namely model checking, to rigorously evaluate the correctness of the generated code.

We evaluate 23 state-of-the-art LLMs on CONCUR, including both proprietary APIs and open-source models. Our framework effectively detects concurrency bugs in LLM-generated programs, achieving 92% precision in manual validation. Furthermore, our analysis reveals that the widely used code evaluation metric, CodeBLEU, fails to reliably reflect correctness of concurrent programs.

The key contributions of our paper are as follows.

- We present a benchmark CONCUR for evaluating the capability of LLMs in concurrent code generation. It consists of 115 concurrency problems paired with structured prompts and validated ground-truth implementations in Java.

- CONCUR also introduces an automated validation framework using formal methods techniques, namely model checking. Unlike existing benchmarks that rely on test cases to assess the correctness of code, CONCUR uses model checking to exhaustively explore the entire state space (within a user-defined bound) of the target concurrent program and verify its correctness.

- We evaluate the capability of 23 LLMs in generating concurrent programs and analyze the potential issues inherent in the concurrent programs produced by these LLMs. The results indicate that our benchmark can effectively identify weaknesses in current LLMs when producing concurrent code.

- We have made our dataset and associated tools publicly available at `https://anonymous.4open.science/r/CONCUR-D205`. We also maintain a public leaderboard at `https://concur-bench.github.io/leaderboard.html`, where we will continue to add new LLMs and expand the benchmark with additional concurrency problems.

## 2 BENCHMARK

To systematically evaluate LLMs on concurrent code generation, we introduce CONCUR, a benchmark that integrates a curated dataset of concurrency problems with a rigorous evaluation framework. The dataset consists of 115 concurrency problems, each paired with structured prompts and validated ground-truth implementations. Building on this foundation, the evaluation framework combines compilation through Java 8 compiler with model checking using Java Pathfinder (JPF) (Havelund & Pressburger, 2000).

### 2.1 DATASET

To evaluate the ability of LLMs to generate concurrent programs, we constructed a dataset of concurrency problems paired with carefully designed prompts and ground-truth implementations. The dataset is designed to ensure that generated outputs can be evaluated consistently and reliably through compilation and model checking.

#### 2.1.1 PROBLEM SELECTION

Two authors systematically reviewed the reference book (Goetz, 2006) and selected concurrency problems to form the basis of the benchmark. Problems that required third-party library packages or could not be validated through back-end output, such as tasks involving database operations or user interface interactions, were excluded. For problems involving a large number of objects, the descriptions were adjusted to reduce object counts. This ensured that LLMs would not produce solutions spawning an excessive number of threads, which would otherwise inflate the state space during model checking and increase execution time. The adjusted problems were then incorporated into the dataset.

Each selected problem was reformulated into a self-contained description suitable for prompting LLMs. The descriptions emphasized concurrency-specific requirements, such as synchronization behavior and inter-thread interactions, while bounding the number of threads and iterations to keep state space exploration tractable. Special care was taken to ensure that these descriptions preserved the spirit of the original problem while also being executable in our evaluation framework. In particular, we explicitly stated input-output behavior and concurrency requirements that might otherwise be implicit in the reference material.

---

Please write a concurrent program in [**Programming language**] that implements the following functionality:
[**Concurrency problem description**].
The requirements are as follows:
[**Specifications**]

---

Figure 1: Prompt for concurrent program generation by LLMs.

### 2.1.2 Prompt engineering

To ensure consistency and reduce noise in generation, we designed a structured prompt (Figure 1) with three components: the programming language, the concurrency problem description, and the specification. In our benchmark, Java 8 was selected as the target language due to its maturity and strong support in analysis tools such as JPF. The framework enforces several constraints: (1) implementations must comply with Java 8 and avoid third-party libraries; (2) thread counts and iterations must be bounded to prevent state explosion during JPF analysis; (3) all code must be consolidated into a single block for compilation; and (4) a public class with a `main` method must be provided as the entry point.

These rules address three practical considerations. First, *concurrent code characteristics:* many problem statements omit explicit object or thread counts, leading LLMs to generate code with excessive threads. Bounding these values mitigates state explosion and improves efficiency. Second, *programming language constraints:* Java 8 prohibits duplicate class names and does not support third-party libraries in our setup. Prompts therefore forbid external imports and enforce unique class definitions to prevent compilation errors. Third, *LLM output behaviors:* LLMs often interleave code with explanations or split solutions across multiple blocks, complicating extraction. To avoid this, prompts explicitly instruct models to output "all classes and methods within a single .java file," ensuring a coherent and compilable program.

### 2.1.3 Ground-truth construction

The reference book provided only incomplete code snippets rather than runnable solutions. To address this, two authors reconstructed complete ground-truth programs by consolidating fragmented code into fully executable implementations. They prepared task lists for each problem, merged them through discussion, and resolved disagreements through consultation with a third author. Each ground-truth solution was extended with a `main` function to guarantee execution of all defined threads and to exercise inter-thread interactions. Thread counts and iteration limits were systematically adjusted to satisfy problem requirements while maintaining computational efficiency. The maximum path execution depth for each program was also determined and recorded as a reference for JPF configuration during evaluation.

### 2.1.4 Mutant generation

To increase the size and diversity of the benchmark, we additionally constructed 72 mutated variants derived from the 43 base problems. These mutations were generated by prompting the Gemini model to produce variations of the original problem descriptions. Each generated mutant was then manually validated to ensure that (1) it preserved the semantics of the original concurrency pattern, (2) it remained compatible with our evaluation framework, and (3) it introduced meaningful structural diversity. We generated three mutants per original problem; after manual validation, we retained 72 semantically correct mutants covering 24 of the 43 base problems. These variations expand the benchmark to 115 total problems and also the mutated descriptions do not appear in the original textbook, reducing the likelihood of evaluation-time memorization.

The final dataset comprises 43 original concurrency problems and 72 validated mutants, resulting in 115 total task instances. The benchmark captures a representative spectrum of concurrency constructs and issues, including `synchronized` blocks, `volatile` variables, the `Lock` interface, `ReentrantLock`, `lock()/unlock()`, `tryLock()`, `Semaphore`, `CountDownLatch`, atomic classes (`Atomic*`), `BlockingQueue`, low-level threading with `Thread` and `Runnable/Callable`, and high-level task management via `ExecutorService`.

### 2.2 Code Evaluation Framework

Figure 2 illustrates the workflow of our evaluation framework. Starting from carefully designed prompts, LLMs generate candidate solutions from which we automatically extract Java code. To assess correctness, our framework combines both compilation in a controlled JDK environment and model checking with JPF. This dual-stage strategy allows us to detect errors that either prevent compilation or emerge only during concurrent execution.

Extracting clean, runnable code from raw LLM outputs is non-trivial, since responses may include interleaved text, explanations, or fragmented snippets. To automate this process, we leverage Java's syntactic structure to identify keywords, encapsulate code into a single coherent file, and prepare it for evaluation. We also account for Java 8 syntax limitations to minimize false negatives caused by environment-specific constraints.

Figure 2: Code Evaluation Framework of CONCUR.

Table 1: Error Types Detected by JPF.

| Error Type | Description |
|---|---|
| Deadlock (DL) | A deadlock occurs when multiple threads are running and each thread waits indefinitely for the others to release their locks. |
| Race Condition (RC) | A race condition occurs when concurrent read and write operations are performed on the same shared variable without proper synchronization. |
| Starvation (SV) | Starvation occurs when one or more threads are perpetually denied access to shared resources because higher-priority threads monopolize execution, preventing the lower-priority threads from making progress. |
| Uncaught Exception (UE) | A range of exceptions, in addition to common concurrency errors such as deadlocks and race conditions, may arise during the execution of concurrent programs, including all exceptions defined in the Java 8 libraries. |
| No Entry Method (NEM) | This error occurs when the generated code lacks a public class with a main method, leaving no appropriate entry point for execution. |
| Single Thread (ST) | This error occurs when the generated code executes with only a single thread. This is considered incorrect, as the prompt explicitly requires the generation of multi-threaded concurrent code. |
| Termination Error (TE) | This error occurs during Java program execution in JPF. These originate not from the program itself but from JPF (e.g., failing to instantiate program variables) or from the runtime environment (e.g., system crashes caused by excessive resource usage from JPF), causing JPF to stop execution. |

### 2.2.1 COMPILATION IN THE JDK ENVIRONMENT

As specified in the prompt framework, all problems must be solved using only default Java 8 libraries, without reliance on third-party packages. After generation, each Java file is named according to the declared `public class` and designated as the entry point for JPF analysis. This step ensures consistency across models and prevents misalignment between file names and class names.

Compilation errors are categorized into three main types:

- Missing package – the program requires an import statement that is absent from the code.
- Syntax error – the program violates Java 8 syntactic rules.
- Third-party dependency – the program attempts to import unsupported external libraries.

To avoid duplicate class name conflicts, only one generated program is compiled and tested at a time. Programs that fail at this stage are labeled as *compilation-failed*. Programs that compile successfully are automatically assigned a JPF configuration file and advanced to the concurrency testing stage, where model checking is performed.

### 2.2.2 MODEL CHECKING WITH JPF

After filtering out programs with compilation errors, further analysis is required to detect potential concurrency bugs. For rigorous validation, we employ explicit-state model checking tool JPF. JPF systematically

explores all possible thread interleavings, traversing execution paths to detect concurrency errors such as deadlocks, race conditions, and other defects listed in Table 1.

JPF uses an extensible listener mechanism that enables fine-grained monitoring of program execution. We leverage default listeners to capture general runtime exceptions and extend them with custom listeners to track properties specific to our study. For example, although a single-threaded program may compile and execute without errors, it fails to satisfy our prompt requirement of generating concurrent code. To address this, we added a listener that reports the number of threads created during execution, allowing us to automatically detect single-threaded outputs. Similarly, we configure listeners to capture starvation and termination errors beyond those caught by default JPF checks.

A key limitation of JPF is the risk of state-space explosion when a program spawns a large number of threads or contains deep branching execution paths. To mitigate this, we bounded the number of threads and iterations in problem descriptions (Section 2.1) and set strict limits in the JPF configuration: the maximum execution depth is capped at ten times that of the corresponding ground truth. Ground-truth solutions were used to calibrate these parameters, ensuring that analysis remains tractable while preserving thorough exploration.

After execution, JPF produces detailed reports, which we parse to detect any errors listed in Table 1. A generated program is considered correct only if it compiles successfully and no errors are reported by JPF.

## 3 Experiments

### 3.1 Experimental setup

The goal of our experiments was to systematically evaluate large language models (LLMs) on the task of concurrent code generation using the CONCUR benchmark. We began by benchmarking 23 LLMs, including both widely used APIs (Anthropic, 2025; OpenAI, 2025) and large open-source models (Yang et al., 2025; Dubey et al., 2024). To ensure comparability, we primarily selected models with more than 30B parameters; in cases where only smaller variants were available, we used the largest publicly released version. Each model was prompted to solve all 115 concurrency problems in our dataset, following the structured framework described in Section 2.2. The outputs were compiled in a controlled Java 8 environment.

To better understand where models fail, we conducted a two-stage error analysis. In the first stage, we examined compilation errors, which were categorized into common causes such as missing imports, syntax violations, and the use of unsupported third-party libraries. In the second stage, we analyzed programs that compiled successfully using Java Pathfinder (JPF). This separation between compilation errors and JPF-detected errors enabled us to distinguish superficial problems of syntactic correctness from deeper issues related to concurrency.

Finally, we investigated how reflective CodeBLEU is as an evaluation metric in this setting. While CodeBLEU has been widely used as a static measure of code similarity, it is unclear whether high CodeBLEU scores correlate with dynamically correct concurrent programs. To address this, we compared CodeBLEU scores against ground-truth implementations and contrasted them with correctness outcomes established through JPF evaluation. This analysis allowed us to assess whether CodeBLEU provides meaningful insights in the context of concurrent code generation or whether dynamic evaluation remains indispensable.

### 3.2 Results

**Benchmarking.** We evaluated the correctness of generated programs using pass@k (Chen et al., 2021), where codes are considered correct if they both compile successfully and pass JPF verification without errors. For each model, we report results under $k \in 1, 3$, meaning that one or three generations per prompt were considered. A JPF configuration file was automatically generated for each Java program that passed compilation, ensuring consistent evaluation against the ground-truth solutions.

Table 2 reveals substantial differences between the pass@1 and pass@3 settings across the 23 evaluated LLMs. Overall, all LLMs achieve higher passing rates under pass@3, demonstrating that the choice of k has a significant impact on the evaluation of concurrent code generation. High-performing LLMs such as gpt-5 and gpt-4o exhibit large improvements (from 77.39% to 91.30% and from 60.00% to 82.61%, respectively). Moreover, 15 LLMs show pass-rate increases exceeding 15 percentage points, indicating a broad sensitivity to multi-sample generation. The effect is even more pronounced among weaker LLMs: codeqwen:7b nearly

Table 2: Comparison of 23 Large Language Models on Concurrent Code Generation for 115 Problems (Pass@1 versus Pass@3).

| Model | Passing Rate | | CodeBLEU | |
|-------|-----|-----|-----|-----|
| | k=1 | k=3 | k=1 | k=3 |
| gpt-5 | 89/115 (77.39%) | 105/115 (91.30%) | 0.576122 | 0.576229 |
| claude-opus-4-1-20250805 | 78/115 (67.83%) | 91/115 (79.13%) | 0.579037 | 0.580484 |
| gemini-3-pro | 76/115 (66.09%) | 85/115 (73.91%) | 0.574317 | 0.575100 |
| gpt-4o | 69/115 (60.00%) | 95/115 (82.61%) | 0.547317 | 0.541937 |
| llama3.3:70b | 57/115 (49.57%) | 71/115 (61.74%) | 0.514043 | 0.514464 |
| qwen3:32b | 53/115 (46.09%) | 75/115 (65.22%) | 0.513917 | 0.509960 |
| wizardcoder:33b | 48/115 (41.74%) | 72/115 (62.61%) | 0.458396 | 0.460315 |
| codestral:22b | 46/115 (40.00%) | 67/115 (58.26%) | 0.469580 | 0.467794 |
| deepseek-r1:32b | 43/115 (37.39%) | 74/115 (64.35%) | 0.533301 | 0.536673 |
| phi4:14b | 39/115 (33.91%) | 68/115 (59.13%) | 0.532162 | 0.531946 |
| phind-codellama:34b | 38/115 (33.04%) | 70/115 (60.87%) | 0.451454 | 0.446808 |
| opencoder:8b | 36/115 (31.30%) | 77/115 (66.96%) | 0.477348 | 0.476352 |
| gemma2:27b | 34/115 (29.57%) | 55/115 (47.83%) | 0.485571 | 0.485590 |
| mixtral:8x7b | 33/115 (28.70%) | 53/115 (46.09%) | 0.471824 | 0.476205 |
| codeqwen:7b | 25/115 (21.74%) | 54/115 (46.96%) | 0.469296 | 0.460804 |
| dolphin3:8b | 22/115 (19.13%) | 44/115 (38.26%) | 0.494357 | 0.490770 |
| magicoder:7b | 12/115 (10.43%) | 27/115 (23.48%) | 0.416781 | 0.412591 |
| llava:34b | 7/115 (6.09%) | 20/115 (17.39%) | 0.451348 | 0.452500 |
| codellama:34b | 7/115 (6.09%) | 23/115 (20.00%) | 0.402659 | 0.397490 |
| vicuna:33b | 6/115 (5.22%) | 20/115 (17.39%) | 0.464678 | 0.452348 |
| starcoder2:15b | 4/115 (3.48%) | 15/115 (13.04%) | 0.357170 | 0.356557 |
| zephyr:7b | 4/115 (3.48%) | 14/115 (12.17%) | 0.530631 | 0.537083 |
| mistral:7b | 3/115 (2.61%) | 14/115 (12.17%) | 0.502507 | 0.502946 |

doubles its accuracy ($21.74\% \rightarrow 46.96\%$), while codellama:34b improves from 6.09% to 20.00%. These findings highlight that allowing multiple generations disproportionately benefits mid- and low-performing LLMs, and that pass@1 alone may substantially underestimate the practical capabilities of many LLMs in concurrent code generation.

To validate the reliability of our automated evaluation, we conducted a manual inspection of LLM-generated code that passed compilation and JPF verification in the pass@1 setting. From the 829 programs identified as correct, we randomly sampled 115 for manual review, ensuring all models and problems are covered. For each selected program, two authors independently constructed a checklist based on the concurrency specification, then merged their assessments through discussion. We examined from the `main` function whether values were transmitted correctly across threads and compared the generated code against the ground truth in terms of classes, functions, data flow, and outputs.

Out of the 115 manually inspected programs, 106 were confirmed correct. Among the remaining cases, 2 exhibited improperly terminated threads, 2 failed to implement thread interaction, 1 contained incomplete class definitions, and 4 passed incorrect variables during execution. Despite these issues, the observed precision of 92% demonstrates that our evaluation framework produces results that are both consistent and reliable, establishing it as a practical benchmark for concurrent code generation.

**Error analysis.** In this experiment, we focused on two categories of errors: compilation errors and JPF-detected errors. At the compilation stage, the primary causes of errors were syntax errors and missing import statements. As illustrated in Figure 3, syntax errors accounted for 71.0% of compilation failures, followed by missing package imports at 26.0%. This finding aligns with observations in sequential code generation (Fan et al., 2023), where syntax errors remain a dominant source of errors. Our results underscore that syntax errors are significant across programming paradigms, including concurrent code generated by LLMs.

Notably, 3.0% of compilation failures stemmed from attempts to use third-party libraries, despite explicit instructions in the prompts to avoid such dependencies. This suggests that some LLMs occasionally prioritize generating seemingly convenient solutions over strictly adhering to task constraints, even when the original intent of the concurrency problems was to solve them using only fundamental language constructs and standard library classes.

Table 3: Summary of Errors in Generated Code (Pass@1)

| Model | Compilation Errors | JPF Errors | | | | | | |
|---|---|---|---|---|---|---|---|---|
| | | DL | RC | SV | UE | NEM | ST | TE |
| claude-opus-4-1-20250805 | 1 | 5 | 7 | 0 | 9 | 0 | 0 | 15 |
| codellama:34b | 88 | 2 | 2 | 0 | 8 | 3 | 2 | 9 |
| codeqwen:7b | 64 | 1 | 3 | 0 | 1 | 6 | 3 | 4 |
| codestral:22b | 42 | 6 | 3 | 0 | 5 | 2 | 1 | 10 |
| deepseek-r1:32b | 43 | 1 | 3 | 0 | 4 | 0 | 1 | 17 |
| dolphin3:8b | 74 | 3 | 2 | 0 | 3 | 2 | 1 | 5 |
| gemini-3-pro | 0 | 6 | 5 | 0 | 14 | 0 | 0 | 3 |
| gemma2:27b | 59 | 5 | 6 | 0 | 3 | 2 | 2 | 4 |
| gpt-4o | 11 | 1 | 7 | 0 | 9 | 0 | 1 | 1 |
| gpt-5 | 2 | 2 | 6 | 0 | 7 | 0 | 1 | 4 |
| llama3.3:70b | 41 | 0 | 9 | 0 | 4 | 2 | 3 | 1 |
| llava:34b | 103 | 2 | 0 | 0 | 0 | 2 | 0 | 1 |
| magicoder:7b | 80 | 3 | 0 | 0 | 0 | 3 | 4 | 8 |
| mistral:7b | 103 | 1 | 1 | 0 | 3 | 2 | 1 | 0 |
| mixtral:8x7b | 66 | 3 | 1 | 0 | 3 | 1 | 3 | 2 |
| opencoder:8b | 43 | 1 | 2 | 0 | 4 | 0 | 1 | 5 |
| phi4:14b | 34 | 2 | 5 | 0 | 5 | 1 | 0 | 6 |
| phind-codellama:34b | 49 | 0 | 1 | 0 | 6 | 3 | 3 | 13 |
| qwen3:32b | 28 | 1 | 6 | 0 | 3 | 2 | 4 | 16 |
| starcoder2:15b | 62 | 0 | 0 | 0 | 0 | 2 | 7 | 1 |
| vicuna:33b | 73 | 0 | 0 | 0 | 1 | 1 | 1 | 1 |
| wizardcoder:33b | 29 | 2 | 2 | 0 | 4 | 1 | 2 | 4 |
| zephyr:7b | 123 | 0 | 0 | 0 | 1 | 0 | 0 | 0 |
| **Total** | **1279** | **68** | **77** | **0** | **214** | **74** | **23** | **81** |

Moving beyond compilation, we analyzed JPF execution logs to categorize concurrency-related errors into the error types mentioned in Table 1. Table 3 summarizes the JPF-reported errors under the pass@1 setting. Among these categories, uncaught exceptions emerged as the most frequent issue, indicating that LLMs often fail to anticipate and handle exceptions during execution. Despite prompts specifying that programs should include a public class and a main function, missing entry points still ranked among the most common errors, alongside termination issues arising from JDK version limitations and the evaluation environment.

Single-threaded execution was also a significant problem. Although these programs technically compiled and executed, they failed to satisfy the concurrency requirements of the tasks. We manually inspected the 23 single-threaded programs identified, and 19 were found to be intended to design for concurrent access to support multi-threaded execution. However, as the generated main thread did not instantiate multiple threads as required, these programs were ultimately classified as incorrect. This reflects a recurring pattern: LLMs often design concurrent logic conceptually but fail to implement the necessary threading constructs to realize actual parallel execution. Through an analysis of the statistics for the concurrency issues, we found that race conditions and deadlocks occurred most frequently, indicating that LLMs are more prone to these types of errors when generating concurrent code. In contrast, starvation was not observed in any case. We hypothesize that this may be due to the relatively low complexity of the code, which allows each thread the opportunity to be scheduled.

A deeper review of JPF reports in the pass@1 setting revealed further limitations. Although JPF is capable of detecting exceptions defined in Java 8 libraries, certain concurrency-related exceptions, such as *IllegalThread-StateException*, were absent from our reports. Two factors account for this: (1) the generated code did not cover the full spectrum of concurrency errors, and (2) LLMs appear reasonably effective at avoiding some of these exceptions when generating code. While our dataset and experiments did not exhaustively exercise every possible concurrency errors, CONCUR nonetheless demonstrates the ability to systematically detect uncaught exceptions supported by JPF.

In addition to the different error categories, we also collected secondary metrics from JPF, including maximum execution depth, total state space, and transition counts. However, we found that these attributes are highly sensitive to minor implementation details and do not correlate reliably with program correctness. Consequently, they are not used as evaluation metrics in our study.

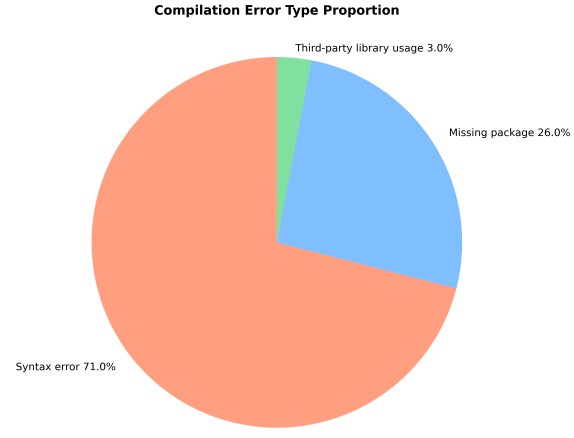

Figure 3: Distribution of compilation errors in 125 randomly selected generated codes.

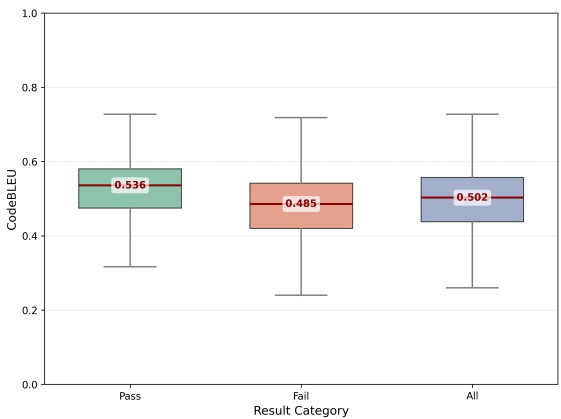

Figure 4: CodeBLEU score ranges for generated code.

**CodeBLEU analysis.** In addition to JPF-based verification, we evaluated the similarity between generated code and ground-truth implementations using CodeBLEU. The original CodeBLEU metric assesses code quality along four dimensions: n-gram overlap, weighted n-gram match, syntax, and data flow. Since our primary interest lies in logical correctness rather than surface-level resemblance, we focus on the syntax and data-flow components.

As shown in Table 2, most LLMs obtain CodeBLEU scores between 0.45 and 0.58 for both k=1 and k=3. High-performing models such as gpt-5, claude-opus-4-1, and phi4:14b reach the upper end of this range, but several models exhibit clear mismatches between CodeBLEU and actual correctness. For example, mistral:7b and mixtral:8x7b achieve moderate CodeBLEU scores (0.47–0.50) despite very low pass rates. This indicates that higher syntax and data-flow similarity does not reliably correspond to correct concurrent behavior. Figure 4 further illustrates this limitation: correct programs have a higher median CodeBLEU (0.536) and a somewhat tighter distribution, yet there is substantial overlap with incorrect programs, some of which even score higher than correct ones. The mean CodeBLEU difference between all generated programs and only passing ones is also minimal, showing that CodeBLEU is only weakly correlated with correctness.

Overall, while CodeBLEU captures certain surface-level similarities, it cannot serve as a standalone metric for evaluating concurrent code generation. Static similarity metrics are insufficient, underscoring the need for dynamic verification tools such as JPF.

## 3.3 Limitations

While Java PathFinder (JPF) is effective in detecting a range of concurrency issues such as race conditions and deadlocks, it cannot capture all possible error types. For example, livelocks and higher-level semantic violations remain undetected. JPF can check for assert violations but only if assertions are present in the code. Further, we run JPF with depth and time bounds, to make the experimentation manageable. As a result, some errors may be missed due to the bound. This restricts the completeness of our concurrency-focused evaluation. Nevertheless, a manual inspection of 115 randomly selected correct programs indicated a recall of 92.2%, suggesting that our approach can be reliable. Furthermore, termination errors (TE) may be inconclusive (e.g., the program may or may not contain an error which might not be discovered due to JPF crashing). However, in this study, we consider this as an error since running JPF on the ground truth solutions completes.

The core problems originate from a widely used textbook, raising the possibility that models were exposed to related material during training. However, the textbook does not provide complete code for most problems; instead, it contains partial snippets that required extensive consolidation, restructuring, and augmentation to form executable programs. This limits the likelihood that an LLM could rely purely on memorization to produce correct solutions.

## 4 RELATED WORK

**LLMs in Code Generation and Benchmarking.** Large language models (LLMs) have become central to software engineering tasks such as code generation (Li et al., 2022; Wang & Chen, 2023; Nejjar et al., 2025), automated program repair (Fan et al., 2023; Joshi et al., 2023; Xia & Zhang, 2022; Zhang et al., 2024; Parasaram et al., 2024; Xia et al., 2023), and code translation (Roziere et al., 2021). Trained on massive code corpora, modern LLMs can synthesize complex structures efficiently, and many models are now widely used as coding assistants in both research and industry (mistral, 2024; Roziere et al., 2023; Bai et al., 2023; Huang et al., 2024; Luo et al., 2023). Despite these advances, their reliability remains limited. LLMs perform well on small, isolated functions but often struggle with end-to-end program generation, where correctness depends on reasoning across multiple components and execution contexts (Li et al., 2022).

To evaluate the capabilities of LLMs in code generation, a variety of benchmarks have been proposed (Du et al., 2024; Chen et al., 2021; Liu et al., 2023; Yu et al., 2024; Austin et al., 2021; Sharma & David, 2025; Athiwaratkun et al., 2022; Lai et al., 2023). These benchmarks typically measure performance using datasets of prompts and ground-truth solutions, allowing comparisons across different models. However, most existing benchmarks emphasize unit- or function-level tasks and therefore do not fully capture the challenges of complex program synthesis. Furthermore, many rely exclusively on static analysis, which is insufficient for uncovering deeper semantic or runtime errors Al Mamun et al. (2010).

A particularly important gap lies in the evaluation of concurrent programs. Current benchmarks overwhelmingly target sequential code (Hendrycks et al., 2021; Li et al., 2022), leaving concurrency underexplored. This is problematic because concurrent programming is pervasive in modern applications, yet testing concurrency introduces significantly greater complexity than testing sequential code (Taylor et al., 1992; Sen, 2007). Compilers can only ensure syntactic correctness, not semantic correctness, and runtime execution alone cannot be relied upon due to nondeterminism in thread scheduling (Carver & Tai, 1998; Fonseca et al., 2011). Thus, existing evaluation strategies are insufficient for benchmarking the generation of reliable multithreaded programs.

**Evaluation for code by static and dynamic analysis** Traditional benchmarks (Du et al., 2024; Chen et al., 2021; Liu et al., 2023; Yu et al., 2024; Austin et al., 2021; Sharma & David, 2025; Athiwaratkun et al., 2022; Lai et al., 2023) employ both static and dynamic analysis. Static analysis often leverages similarity-based metrics, such as BLEU (Papineni et al., 2002), CodeBLEU (Ren et al., 2020), or CrystalBLEU (Eghbali & Pradel, 2022), to compare generated code against references. Compilation is also a necessary prerequisite for both static and dynamic evaluation, ensuring the generated code conforms to language specifications and enabling black-box testing methods such as fuzzing. Dynamic analysis, in turn, executes the generated code under controlled settings to verify functional correctness (Miller et al., 1990).

For concurrent programs, however, these methods are insufficient. Limited runtime execution cannot capture all possible thread interleavings, leaving concurrency-specific issues like race conditions and deadlocks undetected. To address this, we employ Java PathFinder (JPF), a model checker that systematically explores thread schedules. By running generated code under JPF configurations and comparing results to ground truth extracted in advance, our framework enables precise detection of concurrency errors. This provides a rigorous basis for benchmarking LLMs on multithreaded program generation.

## 5 CONCLUSION

We propose a new benchmark, CONCUR, the first benchmark specifically designed to evaluate large language models on concurrent program generation. Unlike existing benchmarks that primarily target sequential code, CONCUR addresses the unique challenges of concurrency by providing a curated dataset of 115 problems and an evaluation framework that integrates compilation with systematic model checking. Through experiments on 23 state-of-the-art LLMs, we show that while many models can produce compilable code, they frequently fail to generate programs that are truly correct under all thread interleavings. Our analysis further demonstrates that static similarity metrics, such as CodeBLEU, do not reliably capture concurrency correctness, underscoring the necessity of the evaluation framework.

In future work, we plan to explore and adopt more concurrent path execution analysis frameworks to extend the benchmark for evaluating code generation across other programming languages.

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

## A  APPENDIX

In this appendix, we provide a detailed description of the procedures for collecting concurrency problems, preparing prompts, and offering more comprehensive information on the evaluation framework.

### A.1  CONCURRENCY PROBLEMS

We conducted a comprehensive review of the book (Goetz, 2006) and systematically screened and selected the code relevant to our study from all chapters. Code segments that depended on third-party libraries, databases, network resources, or front-end components, specifically those whose execution and verification could not be performed solely through back-end processing, were excluded from the analysis. Through this process, we identified 43 distinct concurrency problems that can be addressed exclusively using the basic packages provided in Java 8.

To ensure consistency and minimize selection bias, two authors independently reviewed the book and compiled candidate problems. They then compared their selections, discussed disagreements, and resolved conflicts through joint examination of the code and the concurrency requirements. Final inclusion was based on consensus, ensuring that each problem was both representative of core concurrency concepts and executable under our evaluation framework.

To further guarantee that our ground truth can be executed by JPF with full path exploration and without triggering the errors listed in Table 1, we explicitly define the number of threads for each ground truth. This setting ensures that JPF explores all execution paths within a bounded time and satisfies the concurrency requirements. For example, in Figure 5, a "Thread-safe mutable integer holder" corresponds to a ground truth. The *NUM_THREAD* defined in the ground truth is 2, and in the prompt, we explicitly specify that, in addition to the main thread, the maximum number of threads is 2. We define the upper limit on the number of threads for each prompt in accordance with the requirements of the concurrency problem.

### A.2  PROMPT

When designing the prompt, we divided it into three components: (1) specifying the programming language, (2) describing the concurrency problem, and (3) providing the code specification. As illustrated in Figure 6, this structure accommodates both LLMs that follow fixed output formats and those that do not.

Through preliminary testing, we observed that LLMs are generally capable of consolidating fragmented code snippets into a single cohesive output, and they are less likely to produce multiple alternative solutions when tasked with generating longer code sequences. To ensure consistent and efficient code extraction, we therefore required that all classes and methods be defined within a single `.java` file. When multiple classes are necessary, prompts explicitly instruct models to place them in the same file, with additional nonpublic classes

```
public class ThreadSafeDemo {
  private int value;

  public ThreadSafeDemo(int initialValue) { ...  }
  public ThreadSafeDemo() { ...  }

  public synchronized int get() { ...  }
  public synchronized void set(int value) { ...  }
  public synchronized int increment() { ...  }

  public static void main(String[] args) throws InterruptedException {
    ThreadSafeDemo counter = new ThreadSafeDemo();
    final int NUM_THREADS = 2;
    final int INCREMENTS_PER_THREAD = 2;
    Thread[] threads = new Thread[NUM_THREADS];

    for (int i = 0; i < NUM_THREADS; i++) { ...  }
    for (Thread t :  threads) { t.join(); }

    System.out.println(``Final counter value:  " + counter.get());
    System.out.println(``Expected value:  " + (NUM_THREADS *
INCREMENTS_PER_THREAD));
  }
}
```

Figure 5: Java implementation of a thread-safe mutable integer holder (simplified).

defined using the `class` keyword. This design both simplifies post-processing and reduces the likelihood of extraction errors.

```
Please write a concurrent program in Java 8 that implements the following
functionality:

Create a thread-safe class in Java that holds a mutable integer value.  The
class should have methods to get and set the value, both of which must be
synchronized to ensure safe access and modification from multiple threads.  Use
internal locking (synchronized methods) to protect the integrity of the shared
state.  Additionally, include appropriate annotations to indicate thread-safety
and specify that the internal state is guarded by the object's intrinsic lock.
The requirements are as follows:

All classes and methods must be defined within a single .java file;
The program must include a public static void main(String[] args) method as the
entry point;
No external libraries or dependencies should be used;
The code should be clear, well-structured, and follow Java 8 syntax standards;
If multiple classes are needed, they should all be placed in the same file, with
non-public classes defined using the class keyword;
The implementation must use multithreading to perform tasks concurrently.  You
may use Thread, Runnable, or ExecutorService from the standard Java library to
achieve this.  Ensure that thread safety is considered where necessary;
Write Java 8 code only.  Output a single complete code block using java.  Do
not include any explanations, comments, or extra text.  Only output the code,
nothing else;
The total number of threads used in the program, including the main thread, must
not exceed 3.  Use a fixed thread pool or manually manage at most 2 additional
threads beyond the main thread.  Avoid unbounded thread creation or cached
thread pools;
Do not add any annotations.
```

Figure 6: Example prompt of thread-safe mutable integer holder.

```
target=Main

+vm.fast.startup
listener=gov.nasa.jpf.listener.ThreadCountListener,
gov.nasa.jpf.listener.TimeLimitListener,
gov.nasa.jpf.listener.StarvationListener,
gov.nasa.jpf.listener.PreciseRaceDetector,
gov.nasa.jpf.listener.DeadlockAnalyzer

timeLimitMillis=900000

search.depth_limit = 360
```

Figure 7: Example JPF configuration.

Table 4: List of Uncaught Exceptions Reported by JPF

| Uncaught Exceptions |
| --- |
| gov.nasa.jpf.vm.NoUncaughtExceptionsProperty "java.util.ConcurrentModificationException at java..." |
| gov.nasa.jpf.vm.NoUncaughtExceptionsProperty "java.util.concurrent.RejectedExecutionException: T..." |
| gov.nasa.jpf.vm.NoUncaughtExceptionsProperty "java.util.concurrent.ExecutionException: java.lang..." |
| gov.nasa.jpf.vm.NoUncaughtExceptionsProperty "java.lang.UnsupportedOperationException at java.u..." |
| gov.nasa.jpf.vm.NoUncaughtExceptionsProperty "java.lang.NullPointerException: Calling 'setLocati..." |
| gov.nasa.jpf.vm.NoUncaughtExceptionsProperty "java.lang.NoSuchMethodException: java.lang.String...." |
| gov.nasa.jpf.vm.NoUncaughtExceptionsProperty "java.lang.IndexOutOfBoundsException: toIndex = 150..." |
| gov.nasa.jpf.vm.NoUncaughtExceptionsProperty "java.lang.IllegalArgumentException: Vehicle ID doe..." |
| gov.nasa.jpf.vm.NoUncaughtExceptionsProperty "java.lang.Error: java.lang.NoSuchMethodException: ..." |
| gov.nasa.jpf.vm.NoUncaughtExceptionsProperty "java.lang.Error: java.lang.NoSuchFieldException: t..." |
| gov.nasa.jpf.vm.NoUncaughtExceptionsProperty "java.lang.ClassCastException: java.util.Comparator..." |
| gov.nasa.jpf.vm.NoUncaughtExceptionsProperty "java.lang.AssertionError: java.lang.CloneNotSuppor..." |

### A.3 JPF EVALUATION

Before running JPF verification, a JPF configuration file is created for each successfully compiled Java file. Figure 7 shows an example, where five listeners are employed in JPF verification: (1) ThreadCountListener, which outputs the number of threads created during execution; (2) TimeLimitListener, which forces termination once a preset timeout is reached; (3) StarvationListener, which detects whether any thread remains unscheduled from initiation to termination; (4) PreciseRaceDetector, which identifies potential race conditions by detecting unsynchronized conflicting accesses; and (5) DeadlockAnalyzer, which reports potential deadlocks. The `target` variable specifies the target class to be verified in model checking. `TimeLimitMillis` specified the time bound in seconds (90000s is equal to 15 minutes in this case). `Search.depth_limit` is set to ten times the maximum execution depth of the ground truth, and this restricts the maximum execution depth for JPF.

Figure 8 shows an example of the JPF report, where the detected errors are displayed in the 'results' section. The snapshowt shows the details of the detected error. In addition, JPF also reports the statistics on the path exploration including the number of states, instructions, memory usage, etc.

Table 4 summarizes the uncaught exceptions observed in pass@1. These include both concurrency-specific exceptions and those that may arise in sequential and concurrent programs. Since deadlocks stop the program without raising exceptions, JPF's built-in deadlock detection is essential, while other concurrency issues require additional listeners. For example, starvation is identified using a JPF listener, which classifies a thread as starving if it is never scheduled during execution.

Figure 9 shows a LLM-generated code with a single thread (ST) error. In this case, the program uses Java's `ConcurrentHashMap`, which is inherently thread-safe, but the `main` method only performs initialization and no concurrent operations. As such, the class is designed for multi-threaded scenarios but does not demonstrate concurrency.

Figure 10 illustrates the procedure used for manual validation of generated code against the ground truth. We first check whether the generated program includes the required classes and functions specified in the ground

```
JavaPathfinder core system v8.0 (rev 2345caaa002e15c6147d001d1ccd71e4b538f8e9) - (C) 2005-2014 United States
Government. All rights reserved.

============================================= system under test
VolatileCache.main()

============================================= search started: 25/09/25 9:17 PM

============================================= error 1
gov.nasa.jpf.vm.NotDeadlockedProperty
deadlock encountered:
  thread java.lang.Thread:{id:1,name:Thread-1,status:NEW,priority:5,isDaemon:false,lockCount:0,suspendCount:0}
  thread java.lang.Thread:{id:2,name:Thread-2,status:NEW,priority:5,isDaemon:false,lockCount:0,suspendCount:0}
  thread java.lang.Thread:{id:3,name:pool-1-
thread-1,status:WAITING,priority:5,isDaemon:false,lockCount:0,suspendCount:0}
  thread java.lang.Thread:{id:4,name:pool-1-
thread-2,status:WAITING,priority:5,isDaemon:false,lockCount:0,suspendCount:0}

============================================= snapshot #1
thread java.lang.Thread:{id:3,name:pool-1-
thread-1,status:WAITING,priority:5,isDaemon:false,lockCount:0,suspendCount:0}
  waiting on: java.lang.Thread$Permit@2cc
  call stack:
      at sun.misc.Unsafe.park(Unsafe.java)
      at java.util.concurrent.locks.LockSupport.park(LockSupport.java:175)
      at
java.util.concurrent.locks.AbstractQueuedSynchronizer$ConditionObject.await(AbstractQueuedSynchronizer.java:2039)
      at java.util.concurrent.LinkedBlockingQueue.take(LinkedBlockingQueue.java:442)
      at java.util.concurrent.ThreadPoolExecutor.getTask(ThreadPoolExecutor.java:1074)
      at java.util.concurrent.ThreadPoolExecutor.runWorker(ThreadPoolExecutor.java:1134)
      at java.util.concurrent.ThreadPoolExecutor$Worker.run(ThreadPoolExecutor.java:624)

thread java.lang.Thread:{id:4,name:pool-1-
thread-2,status:WAITING,priority:5,isDaemon:false,lockCount:0,suspendCount:0}
  waiting on: java.lang.Thread$Permit@2de
  call stack:
      at sun.misc.Unsafe.park(Unsafe.java)
      at java.util.concurrent.locks.LockSupport.park(LockSupport.java:175)
      at
java.util.concurrent.locks.AbstractQueuedSynchronizer$ConditionObject.await(AbstractQueuedSynchronizer.java:2039)
      at java.util.concurrent.LinkedBlockingQueue.take(LinkedBlockingQueue.java:442)
      at java.util.concurrent.ThreadPoolExecutor.getTask(ThreadPoolExecutor.java:1074)
      at java.util.concurrent.ThreadPoolExecutor.runWorker(ThreadPoolExecutor.java:1134)
      at java.util.concurrent.ThreadPoolExecutor$Worker.run(ThreadPoolExecutor.java:624)

============================================= thread ops #1
   4       3     trans    insn       loc              : stmt
------- ------- ---------------------------------------------------
W:2de   |         25      invokevirtual java/util/concurrent/locks/LockSupport.java:175
  |     W:2cc     24      invokevirtual java/util/concurrent/locks/LockSupport.java:175
  S     |          2          runstart java/util/concurrent/ThreadPoolExecutor.java:0
  |     S          0          runstart java/util/concurrent/ThreadPoolExecutor.java:0
Unique logical threads created during execution: 3

============================================= results
error #1: gov.nasa.jpf.vm.NotDeadlockedProperty "deadlock encountered:    thread java.lang.Thread:{i..."

============================================= statistics
elapsed time:        00:00:00
states:              new=26,visited=0,backtracked=0,end=1
search:              maxDepth=26,constraints=0
choice generators:   thread=26 (signal=0,lock=9,sharedRef=13,threadApi=2,reschedule=1), data=0
heap:                new=1205,released=97,maxLive=1104,gcCycles=26
instructions:        19441
max memory:          245MB
loaded code:         classes=135,methods=2802

============================================= search finished: 25/09/25 9:17 PM
```

Figure 8: JPF Report.

truth. If these elements are present, we then examine the `main` function to verify that they are invoked and used correctly according to the prompt. To support this process, we systematically analyze each prompt and construct a to-do list that captures the essential concurrency requirements expected in the code.

For example, the following to-do list corresponds to the prompt shown in Figure 10:

```
import java.util.*;
import java.util.concurrent.*;
import java.lang.ref.*;

public class DelegatingVehicleTracker {
  private final ConcurrentHashMap<String, String> locations = new
ConcurrentHashMap<>();

  public DelegatingVehicleTracker(Map<String, String> initialLocations) {
    this.locations.putAll(initialLocations);
  }

  public Optional<String> getLocation(String vehicleId) {
    return Optional.ofNullable(this.locations.get(vehicleId));
  }

  public boolean setLocation(String vehicleId, String location) {
    if (!this.locations.containsKey(vehicleId)) {
      return false;
    } else {
      this.locations.put(vehicleId, location);
      return true;
    }
  }

  public Map<String, String> getAllLocations() {
    return Collections.unmodifiableMap(new HashMap<>(this.locations));
  }

  public static void main(String[] args) throws Exception {
    Map<String, String> initialLocations = new HashMap<>();
    initialLocations.put(''1", ''San Francisco");
    initialLocations.put(''2", ''New York");
    DelegatingVehicleTracker tracker = new
DelegatingVehicleTracker(initialLocations);
  }
}
```

Figure 9: Java implementation of a thread-safe vehicle tracker using ConcurrentHashMap. This code ensures concurrency safety; however, it does not constitute a concurrent program. Instead, it functions as a thread-safe abstraction for data access.

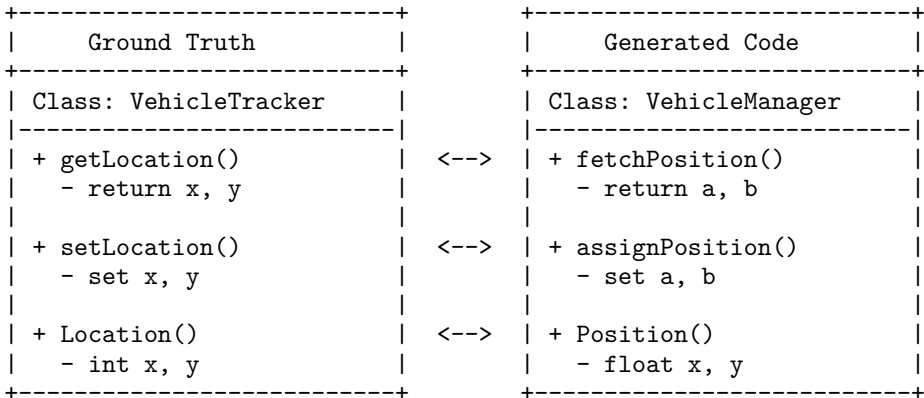

Figure 10: Class and function comparison between ground truth and generated code.

- Generate two threads.

- One thread sets the location, while the other thread gets the location.

- Use a synchronized method to ensure thread safety.

Figure 11 presents an example of incorrect code identified during manual inspection. Although the program compiles successfully and shows no immediate errors, the two threads created in the `main` function operate independently without sharing resources. Each thread instantiates its own object of `ServerStatusBeforeSplit`, rather than sharing a common instance. Consequently, the program fails to satisfy the prompt's requirement for thread interaction.

```
import java.util.HashSet;
import java.util.concurrent.ExecutorService;
import java.util.concurrent.Executors;
import java.util.concurrent.ThreadPoolExecutor;
import java.util.concurrent.TimeUnit;

public class ServerStatusBeforeSplit {
  private HashSet<String> users;
  private HashSet<String> queries;
  public ServerStatusBeforeSplit() {
    users = new HashSet<>();
    queries = new HashSet<>();
  }

  }

...
  public static void main(String[] args) {
    Thread t1 = new Thread(new Runnable() {
      @Override
      public void run() {
        for (int i = 0; i < 2; i++) {
          try {
            TimeUnit.MILLISECONDS.sleep(5);
          } catch (InterruptedException e) {
            e.printStackTrace();
          }
          ServerStatusBeforeSplit ssb = new ServerStatusBeforeSplit();
...           }
      }
    });
    Thread t2 = new Thread(new Runnable() {
      @Override
      public void run() {
        for (int i = 0; i < 2; i++) {
          try {
            TimeUnit.MILLISECONDS.sleep(5);
          } catch (InterruptedException e) {
            e.printStackTrace();
          }
          ServerStatusBeforeSplit ssb = new ServerStatusBeforeSplit();
...           }
      }
    });
    ThreadPoolExecutor executor = (ThreadPoolExecutor)
Executors.newFixedThreadPool(2);
    executor.submit(t1);
    executor.submit(t2);
    executor.shutdown();
  }
}
```

Figure 11: Java code generated by LLM-implementation of `ServerStatusBeforeSplit`. Although this program compiles and runs without errors, the two threads do not share resources or interact, and thus it does not meet the prompt's requirement for thread interaction.

