# OpenReview forum: "CONCUR: Benchmarking LLMs for Concurrent Code Generation"
_ICLR.cc/2026/Conference — Submitted to ICLR 2026_

### Official Review · Reviewer_fAkt · 2025-10-19

**Soundness:** 2
**Presentation:** 3
**Contribution:** 2
**Rating:** 2
**Confidence:** 4

**Summary:**

This paper introduces CONCUR, the first benchmark specifically designed to evaluate large language models on concurrent program generation — a domain much more complex than sequential code due to non-deterministic thread scheduling, synchronization, and concurrency-specific bugs (e.g., deadlocks, race conditions, starvation).
The benchmark includes 43 curated Java concurrency problems derived from Java Concurrency in Practice (Goetz, 2006), each with structured prompts and verified ground-truth solutions.

CONCUR employs formal verification via Java Pathfinder for exhaustive state-space exploration, detecting concurrency issues beyond what static metrics or unit tests can capture. The authors evaluate 22 state-of-the-art LLMs (including GPT-5, Claude-Opus-4.1, GPT-4o, Qwen-3, DeepSeek-R1, etc.) and demonstrate that while LLMs can often produce compilable code, they frequently fail under full concurrency verification.
The study further shows that static similarity metrics like CodeBLEU fail to correlate with true correctness, emphasizing the need for formal, dynamic evaluation frameworks.

**Strengths:**

- Novel Benchmark Domain

Comprehensive benchmark targeting concurrent code generation, addressing a crucial but overlooked area in code intelligence research.

- Formal Verification Integration

The use of model checking introduces rigor, enabling the detection of deep concurrency bugs (deadlocks, race conditions) that conventional test-based evaluation misses.

- Comprehensive Empirical Evaluation

Evaluates 22 diverse LLMs under uniform conditions, includes manual validation, and provides a public leaderboard and dataset.

**Weaknesses:**

- Limited Dataset Scale and Diversity

Only 43 problems, all from a single Java text book, limit coverage and generalization to broader concurrency paradigms (e.g., message passing, lock-free, or distributed models).

- Single-Language Restriction (Java 8)

The benchmark excludes other major concurrency ecosystems like C++, Go, or Rust, reducing cross-language insight.

- Partial Coverage of Concurrency Semantics

JPF bounds and model-checking limitations (e.g., no livelock detection, time depth cutoffs) may miss certain concurrency issues, limiting completeness of the evaluation.

- Evaluation granularity

The analysis could benefit from qualitative insights into why models fail (e.g., incorrect synchronization pattern, wrong locking scope).
No ablation on prompt variants or temperature settings.

**Questions:**

– Do the authors plan to extend CONCUR to multiple programming languages (e.g., Go, Rust) that have different concurrency models? This would broaden its applicability and reveal model generalization across paradigms.

– Since JPF focuses on low-level interleavings, have the authors considered incorporating semantic invariants or assertion checking to detect higher-level correctness violations (e.g., protocol violations, data consistency)?

---

> ### Author Response · Authors · 2025-11-20
> **Response to the comments 1-3 of Reviewer fAkt**
>
> ### Comment 1: Limited Dataset Scale and Diversity
> Only 43 problems, all from a single Java text book, limit coverage and generalization to broader concurrency paradigms (e.g., message passing, lock-free, or distributed models).
>
> ### Response:
> We appreciate the reviewer’s concern regarding the number of benchmark programs. While the core benchmark consists of 43 real concurrency problems, we have additionally constructed 72 mutated variants derived from these base problems to increase the size of the benchmark. These mutations were built by leveraging Gemini prompted to generate variations of the problem descriptions, which we validated manually, resulting in a benchmark of 115 total problems. We generated three mutants for each original problem, and after manual validation we ended up with meaningful mutants for 24 problems without changing their original semantics.
>
> Also,note that  the problem set provides broad coverage of known multithreaded programming patterns and pitfalls. Across the original and mutated problems, the benchmark captures 13 representative categories of concurrency constructs and issues:
>
> - synchronized
> - volatile
> - Lock interface
> - ReentrantLock
> - lock() / unlock()
> - tryLock()
> - Semaphore
> - CountDownLatch
> - Atomic* classes
> - BlockingQueue
> - Thread
> - Runnable / Callable
> - ExecutorService
>
> This expanded benchmark allows us to systematically test LLM behavior across a diverse and realistic spectrum of multithreaded scenarios. Thus, although the number of original source programs is 43, the final benchmark includes 115 problems, offering a comprehensive evaluation of concurrent programming capabilities.  The ground-truth implementations range from 35–140 lines of code and 2–30 functions, providing substantially more complexity than simple textbook snippets.
>
> To contextualize the scale, widely used code-generation benchmarks such as HumanEval include around 160 problems, showing that benchmarks of this magnitude are common and effective in practice. Given the substantially higher complexity of multithreaded programming,we believe a dataset of 115 problems represents a good size for an initial benchmark in this domain.
>
> Regarding contamination risk, the benchmark programs are not direct copies of the textbook examples. Many snippets were incomplete and required significant supplementation; we also adjusted thread counts and concurrency structures, and verified all programs using Java PathFinder (JPF). The 72 mutated problems further introduce concurrency variations that do not appear in the textbook, reducing the likelihood of memorization.
>
> We will include a discussion of potential contamination in the paper. Overall, despite using textbook concepts as a starting point, the final benchmark is more diverse and structurally modified, providing meaningful coverage for evaluating LLMs on concurrent programming.
>
>
> ### Comment 2: Single-Language Restriction (Java 8)
> The benchmark excludes other major concurrency ecosystems like C++, Go, or Rust, reducing cross-language insight.
> Do the authors plan to extend CONCUR to multiple programming languages (e.g., Go, Rust) that have different concurrency models? This would broaden its applicability and reveal model generalization across paradigms.
>
> ### Response:
> We agree that extending the benchmark beyond Java, such as incorporating languages with stronger static concurrency guarantees like Go, Rust, would provide additional valuable perspectives. We chose Java for the first release of our benchmark because it has been widely used for concurrent programming and has strong support from existing mature model checking tools. Expanding the benchmark to other programming languages requires mature model checking tool support and we leave this for future work.
>
> ### Comment 3: Partial Coverage of Concurrency Semantics
> JPF bounds and model-checking limitations (e.g., no livelock detection, time depth cutoffs) may miss certain concurrency issues, limiting completeness of the evaluation.
>
> ### Response:
> We appreciate the reviewer’s concern regarding the inherent limitations of model checking. We agree that JPF’s bounds and exploration limits can in principle miss certain concurrency issues. However, we selected JPF because it remains one of the most rigorous and systematic tools available for exploring thread interleavings, races, and deadlocks in Java programs.
>
> For our benchmark, JPF provides a substantially stronger guarantee than traditional testing, which cannot feasibly capture nondeterministic schedules across 115 programs and 13 categories of concurrency constructs. Nonetheless, we acknowledge that no verification method is perfectly complete. We will add a discussion in the revised paper outlining these limitations and clarifying the scope of what JPF can and cannot detect within our evaluation framework.

---

> > ### Author Response · Authors · 2025-11-20
> > **Response to the comment 4 and the question of Reviewer fAkt**
> >
> > ### Comment 4: Evaluation granularity
> > The analysis could benefit from qualitative insights into why models fail (e.g., incorrect synchronization pattern, wrong locking scope). No ablation on prompt variants or temperature settings.
> >
> > ### Response:
> > We appreciate the reviewer’s suggestion. We agree that qualitative insights would strengthen the analysis. In the revised paper, we will conduct a qualitative examination of model failures, including identifying which types of concurrency problems (across the 13 categories) are most frequently failed by LLMs and analyzing the specific types of mistakes they make (e.g., incorrect synchronization patterns, improper locking scopes, misuse of concurrency primitives). We will report these findings in the updated version of the paper.
> >
> >
> > ### Q1: Since JPF focuses on low-level interleavings, have the authors considered incorporating semantic invariants or assertion checking to detect higher-level correctness violations (e.g., protocol violations, data consistency)?
> >
> > ### Response:
> > Adding properties to check (in the form of assertions, invariants, or monitors, which JPF can check) would make the evaluation even stronger. In future work we plan to investigate using LLMs to generate such properties from the natural language description (a challenging research area in itself).

---

> > > ### Author Response · Authors · 2025-11-21
> > > **Result of the updated experiments**
> > >
> > > The table reports Pass@1 accuracy for four prompt categories: (1) Original 43 Problems, which represent the full set included in prior evaluation; (2) 24 Mutation-Eligible Problems, the subset selected for prompt mutation; (3) 72 Mutated Problems, generated from that subset; and (4) All 115 Problems, combining original and mutated versions. This structure allows comparison across the different prompt groups and highlights how models behave under each configuration.
> > >
> > > Across the 22 LLMs evaluated, the results show a balanced outcome. Some models—such as gpt-5, claude-opus-4-1, and gpt-4o—obtain higher pass rates on the mutated prompts, whereas others—such as qwen3-32b, wizardcoder-33b, and phi-4-14b—perform better on the original prompts. This distribution suggests that the mutated prompts preserve the overall difficulty profile while providing additional variation. The combined Pass@1 results over all 115 prompts also follow consistent trends across models.
> > >
> > >
> > > ## LLM Pass@1 Rate Comparison
> > >
> > > | LLM | Original 43 Problems | 24 Mutation-Eligible Problems | 72 Mutated Problems | All 115 Problems |
> > > | :--- | :--- | :--- | :--- | :--- |
> > > | gpt-5 | 72.09% (31/43) | 83.33% (20/24) | 79.17% (57/72) | 76.52% (88/115) |
> > > | claude-opus-4-1 | 62.79% (27/43) | 58.33% (14/24) | 70.83% (51/72) | 67.83% (78/115) |
> > > | gpt-4o | 53.49% (23/43) | 50.00% (12/24) | 68.06% (49/72) | 62.61% (72/115) |
> > > | llama3.3:70b | 46.51% (20/43) | 45.83% (11/24) | 48.61% (35/72) | 47.83% (55/115) |
> > > | qwen3:32b | 41.86% (18/43) | 62.50% (15/24) | 41.67% (30/72) | 41.74% (48/115) |
> > > | codestral:22b | 37.21% (16/43) | 29.17% (7/24) | 44.44% (32/72) | 41.74% (48/115) |
> > > | wizardcoder:33b | 34.88% (15/43) | 41.67% (10/24) | 37.50% (27/72) | 36.52% (42/115) |
> > > | phi4:14b | 44.19% (19/43) | 37.50% (9/24) | 27.78% (20/72) | 33.91% (39/115) |
> > > | deepseek-r1:32b | 27.91% (12/43) | 29.17% (7/24) | 37.50% (27/72) | 33.91% (39/115) |
> > > | phind-codellama:34b | 25.58% (11/43) | 41.67% (10/24) | 37.50% (27/72) | 33.04% (38/115) |
> > > | gemma2:27b | 27.91% (12/43) | 16.67% (4/24) | 34.72% (25/72) | 32.17% (37/115) |
> > > | opencoder:8b | 34.88% (15/43) | 33.33% (8/24) | 29.17% (21/72) | 31.30% (36/115) |
> > > | mixtral:8x7b | 30.23% (13/43) | 25.00% (6/24) | 26.39% (19/72) | 27.83% (32/115) |
> > > | codeqwen:7b | 13.95% (6/43) | 12.50% (3/24) | 26.39% (19/72) | 21.74% (25/115) |
> > > | dolphin3:8b | 23.26% (10/43) | 20.83% (5/24) | 18.06% (13/72) | 20.00% (23/115) |
> > > | magicoder:7b | 16.28% (7/43) | 4.17% (1/24) | 11.11% (8/72) | 13.04% (15/115) |
> > > | vicuna:33b | 11.63% (5/43) | 8.33% (2/24) | 5.56% (4/72) | 7.83% (9/115) |
> > > | llava:34b | 9.30% (4/43) | 16.67% (4/24) | 4.17% (3/72) | 6.09% (7/115) |
> > > | starcoder2:15b | 6.98% (3/43) | 12.50% (3/24) | 0.00% (0/72) | 2.61% (3/115) |
> > > | zephyr:7b | 4.65% (2/43) | 4.17% (1/24) | 2.78% (2/72) | 3.48% (4/115) |
> > > | mistral:7b | 4.65% (2/43) | 8.33% (2/24) | 2.78% (2/72) | 3.48% (4/115) |
> > > | codellama:34b | 0.00% (0/43) | 0.00% (0/24) | 2.78% (2/72) | 1.74% (2/115) |
> > >
> > > ---

---

### Official Review · Reviewer_DzG4 · 2025-10-31

**Soundness:** 4
**Presentation:** 3
**Contribution:** 3
**Rating:** 8
**Confidence:** 4

**Summary:**

The authors develop a new benchmark for concurrent programming, which consists of 43 curated concurrency problems drawn from a textbook, with structured prompts and ground-truth implementations in Java.  The authors evaluate 22 LLMs on the benchmark.

Unlike most coding benchmarks, success is not determined by the ability to pass unit tests, because concurrent programming bugs like race conditions and deadlocks are notoriously difficult to catch with unit tests.  Instead, the authors perform model-checking with Java Pathfinder (JPF), to find concurrency-related bugs in the output code.

Somewhat surprisingly, the vast majority of errors are still compilation errors, with the majority being syntax errors.  LLMs still struggle to write syntactically valid code.   A second surprise is that LLMs often implement conceptually correct code, but then fail to actually spawn multiple threads to execute it.

The authors find that CodeBLEU is a poor metric for code quality.

**Strengths:**

The benchmark seems to be well-designed, and the evaluation of the 22 LLMs is thorough, covering all of the important models.

By far the greatest strength of this benchmark is its use of model-checking to catch concurrency bugs.  As the use of coding LLMs continues to proliferate, so will LLM-introduced bugs, and concurrency-related bugs like race conditions are notoriously difficult to catch. Formal static or dynamic analysis is currently severely underused as a way to evaluate code quality, so I will champion this paper as an important milestone in teaching LLMs to write correct code, by using formal measures of code correctness.  Hopefully future work will follow the same path.

**Weaknesses:**

The benchmark only contains 43 problems.

Perhaps most importantly the problems are all drawn from a textbook, which was published almost 20 years ago.  This means that the textbook, or similar problems, are likely in the training data of SOTA LLMs.

The benchmark would be strengthened by having more problems, and including not-previously published problems.  It would be interesting to include problems in a language other than Java -- e.g. the Rust type system also protects against concurrency bugs.

**Questions:**

None.

---

> ### Author Response · Authors · 2025-11-20
> **Response to the comments of Reviewer DzG4**
>
> ### Comment 1: The benchmark only contains 43 problems.
> The benchmark would be strengthened by having more problems, and including not-previously published problems. It would be interesting to include problems in a language other than Java -- e.g. the Rust type system also protects against concurrency bugs.
> ### Comment 2: Perhaps most importantly the problems are all drawn from a textbook, which was published almost 20 years ago. This means that the textbook, or similar problems, are likely in the training data of SOTA LLMs.
>
> ### Response:
> We thank the reviewer for the thoughtful and supportive review.
>
> We agree that the number of our benchmark programs is limited, and expanding it with more problems is valuable. We chose Java because it has been widely used for concurrent programming and strong support from existing mature model checking tools. Expanding the benchmark to other programming languages requires mature model checking tool support and we leave this for future work.
>
> Our goal in this initial release was to ensure high-quality verified problems, accompanied by ground-truth implementations and model-checking results, which is non-trivial to construct. We prioritized quality over quantity in this first version.  This problem set provides broad coverage of known concurrent programming patterns and pitfalls. Across the original and mutated programs, the benchmark captures 13 representative categories of concurrency constructs and issues:
>
> - synchronized
> - volatile
> - Lock interface
> - ReentrantLock
> - lock() / unlock()
> - tryLock()
> - Semaphore
> - CountDownLatch
> - Atomic* classes
> - BlockingQueue
> - Thread
> - Runnable / Callable
> - ExecutorService
>
> According to your suggestion, we have now expanded the benchmark to 115 problems. We have now additionally constructed 72 mutated variants derived from these base problems to increase the size of the benchmark. These mutations were built by leveraging Gemini prompted to generate variations of the problem descriptions. We generated three mutants for each original problem, and after manual validation we ended up with 72 meaningful mutants for 24 problems without changing their original semantics. We hope this expanded benchmark can allow us to systematically evaluate LLM capabilities across a diverse and realistic spectrum of concurrent code scenarios.
>
> We acknowledge the concern that the problems from the textbook are likely in the training data of SOTA LLMs. That said, most problems do not have complete code solutions. Instead, the code snippets provided in the textbook were incomplete and required significant supplementation and restructuring. This may partially explain why many LLMs are still struggling to generate correct code for these concurrency problems as presented in our paper. The 72 mutated problems further introduce concurrency variations that do not appear in the textbook, reducing the likelihood of memorization. We will include a discussion of potential contamination in the paper. Overall, despite using textbook concepts as a starting point, the final benchmark is more diverse, and structurally modified, providing meaningful coverage for evaluating LLMs on concurrent programming.

---

> > ### Author Response · Authors · 2025-11-21
> > **Result of the updated experiments**
> >
> > The table reports Pass@1 accuracy for four prompt categories: (1) Original 43 Problems, which represent the full set included in prior evaluation; (2) 24 Mutation-Eligible Problems, the subset selected for prompt mutation; (3) 72 Mutated Problems, generated from that subset; and (4) All 115 Problems, combining original and mutated versions. This structure allows comparison across the different prompt groups and highlights how models behave under each configuration.
> >
> > Across the 22 LLMs evaluated, the results show a balanced outcome. Some models—such as gpt-5, claude-opus-4-1, and gpt-4o—obtain higher pass rates on the mutated prompts, whereas others—such as qwen3-32b, wizardcoder-33b, and phi-4-14b—perform better on the original prompts. This distribution suggests that the mutated prompts preserve the overall difficulty profile while providing additional variation. The combined Pass@1 results over all 115 prompts also follow consistent trends across models.
> >
> >
> > ## LLM Pass@1 Rate Comparison
> >
> > | LLM | Original 43 Problems | 24 Mutation-Eligible Problems | 72 Mutated Problems | All 115 Problems |
> > | :--- | :--- | :--- | :--- | :--- |
> > | gpt-5 | 72.09% (31/43) | 83.33% (20/24) | 79.17% (57/72) | 76.52% (88/115) |
> > | claude-opus-4-1 | 62.79% (27/43) | 58.33% (14/24) | 70.83% (51/72) | 67.83% (78/115) |
> > | gpt-4o | 53.49% (23/43) | 50.00% (12/24) | 68.06% (49/72) | 62.61% (72/115) |
> > | llama3.3:70b | 46.51% (20/43) | 45.83% (11/24) | 48.61% (35/72) | 47.83% (55/115) |
> > | qwen3:32b | 41.86% (18/43) | 62.50% (15/24) | 41.67% (30/72) | 41.74% (48/115) |
> > | codestral:22b | 37.21% (16/43) | 29.17% (7/24) | 44.44% (32/72) | 41.74% (48/115) |
> > | wizardcoder:33b | 34.88% (15/43) | 41.67% (10/24) | 37.50% (27/72) | 36.52% (42/115) |
> > | phi4:14b | 44.19% (19/43) | 37.50% (9/24) | 27.78% (20/72) | 33.91% (39/115) |
> > | deepseek-r1:32b | 27.91% (12/43) | 29.17% (7/24) | 37.50% (27/72) | 33.91% (39/115) |
> > | phind-codellama:34b | 25.58% (11/43) | 41.67% (10/24) | 37.50% (27/72) | 33.04% (38/115) |
> > | gemma2:27b | 27.91% (12/43) | 16.67% (4/24) | 34.72% (25/72) | 32.17% (37/115) |
> > | opencoder:8b | 34.88% (15/43) | 33.33% (8/24) | 29.17% (21/72) | 31.30% (36/115) |
> > | mixtral:8x7b | 30.23% (13/43) | 25.00% (6/24) | 26.39% (19/72) | 27.83% (32/115) |
> > | codeqwen:7b | 13.95% (6/43) | 12.50% (3/24) | 26.39% (19/72) | 21.74% (25/115) |
> > | dolphin3:8b | 23.26% (10/43) | 20.83% (5/24) | 18.06% (13/72) | 20.00% (23/115) |
> > | magicoder:7b | 16.28% (7/43) | 4.17% (1/24) | 11.11% (8/72) | 13.04% (15/115) |
> > | vicuna:33b | 11.63% (5/43) | 8.33% (2/24) | 5.56% (4/72) | 7.83% (9/115) |
> > | llava:34b | 9.30% (4/43) | 16.67% (4/24) | 4.17% (3/72) | 6.09% (7/115) |
> > | starcoder2:15b | 6.98% (3/43) | 12.50% (3/24) | 0.00% (0/72) | 2.61% (3/115) |
> > | zephyr:7b | 4.65% (2/43) | 4.17% (1/24) | 2.78% (2/72) | 3.48% (4/115) |
> > | mistral:7b | 4.65% (2/43) | 8.33% (2/24) | 2.78% (2/72) | 3.48% (4/115) |
> > | codellama:34b | 0.00% (0/43) | 0.00% (0/24) | 2.78% (2/72) | 1.74% (2/115) |
> >
> > ---

---

> > ### Comment · Reviewer_DzG4 · 2025-11-21
> >
> > Good to hear, I think that strengthens the benchmark.  Would it be at all possible to get gemini into your result list, for comparison against GPT and claude?

---

> ### Author Response · Authors · 2025-11-22
> **Results of Gemini**
>
> Thank you for the suggestion. Passing rate of "gemini-3-pro":
>
> For original 43 problems: 62.79% (27/43)
>
> For mutation eligible 24 problems: 70.83% (17/24)
>
> For mutated 72 problems: 68.06% (49/72)
>
> For all 115 problems: 66.09% (76/115)

---

> > ### Comment · Reviewer_DzG4 · 2025-11-22
> >
> > Thank you!  That's an important point of comparison, especially since Gemini 3 is supposed to be better at coding...

---

### Official Review · Reviewer_oHT5 · 2025-10-31

**Soundness:** 3
**Presentation:** 3
**Contribution:** 2
**Rating:** 2
**Confidence:** 4

**Summary:**

This paper introduces a benchmark that measures how well code generation models are able to generate concurrent Java code. The authors construct a small problem set sourced from a Java concurrency textbook and measure success based on compilation success and Java PathFinder (JPF) verification.

**Strengths:**

- Paper is well-written and motivations are sound (most code benchmarks focus on single proc. code).
- Thoughtful methodology. Steps are taken to ensure solutions can be verified without excessive computation resources (e.g. by limiting num threads, etc)
- Paper shows benefit of going beyond CodeBLEU and provides error analysis of top models.

**Weaknesses:**

- Benchmark is simple and problems are sourced from a textbook published in 2006. The proposed test set is only 43 problems, curated by the authors, which is extremely small. Additionally there is a risk of contamination as models may have trained on this textbook. The authors do not provide any insight into the contamination risk.
- Evaluation metric is based on compilation and verification success, not test case passing. Due to the nature of the problems (which are presented without completed solutions in the original data source), the authors resort to compilation and JPF based checks to measure success. However, the best verification for code is running against a test suite as code that can compile and pass JPF checks could still be wrong.
- Benchmark initial performance is quite high-- GPT-5 is at 72% pass@1.

**Questions:**

1) Can the authors explain any experiments or measures taken to avoid contamination given that the textbook may be in the training set of these LLMs?
2) Is the proposed test set (just 43 problems) a statistically significant sample size for estimating concurrent code generation capabilities?
3) Why is compilation success + JPF verification a sufficient measure of performance (instead of say compilation + JPF + running code against test suite)?

---

> ### Author Response · Authors · 2025-11-20
> **Response to the comments of Reviewer oHT5**
>
> ### Comment 1: Benchmark is simple and problems are sourced from a textbook published in 2006. The proposed test set is only 43 problems, curated by the authors, which is extremely small. Additionally there is a risk of contamination as models may have trained on this textbook. The authors do not provide any insight into the contamination risk.
> ### Q1: Can the authors explain any experiments or measures taken to avoid contamination given that the textbook may be in the training set of these LLMs?
>
> ### Response:
> We appreciate the reviewer’s concern regarding the number of benchmark programs. While the core benchmark consists of 43 concurrency problems, we have additionally constructed 72 mutated variants derived from these base problems to increase the size of the benchmark. These mutations were built by leveraging Gemini prompted to generate variations of the problem descriptions, which we validated manually, resulting in a benchmark of 115 total problems. We generated three mutants for each original problem, and after manual validation we ended up with meaningful mutants for 24 problems without changing their original semantics.
>
> Also,note that  the problem set provides broad coverage of known multithreaded programming patterns and pitfalls. Across the original and mutated programs, the benchmark captures 13 representative categories of concurrency constructs and issues:
>
> - synchronized
> - volatile
> - Lock interface
> - ReentrantLock
> - lock() / unlock()
> - tryLock()
> - Semaphore
> - CountDownLatch
> - Atomic* classes
> - BlockingQueue
> - Thread
> - Runnable / Callable
> - ExecutorService
>
> This expanded benchmark allows us to systematically test LLM behavior across a diverse and realistic spectrum of multithreaded scenarios. Thus, although the number of original source programs is 43, the final benchmark includes 115 problems, offering a comprehensive evaluation of concurrent programming capabilities.  The ground-truth implementations range from 35–140 lines of code and 2–30 functions, providing substantially more complexity than simple textbook snippets.
>
> Regarding contamination risk, the benchmark programs are not direct copies of the textbook examples. Many snippets were incomplete and required significant supplementation; we also adjusted thread counts and concurrency structures, and verified all programs using Java PathFinder (JPF). The 72 mutated problems further introduce concurrency variations that do not appear in the textbook, reducing the likelihood of memorization.
>
> We will include a discussion of potential contamination in the paper. Overall, despite using textbook concepts as a starting point, we believe that the final benchmark is more diverse, and structurally modified, providing meaningful coverage for evaluating LLMs on concurrent programming.
>
> To contextualize the scale, widely used code-generation benchmarks such as HumanEval include around 160 problems, showing that benchmarks of this magnitude are common and effective in practice. Given the substantially higher complexity of multithreaded programming,we believe a dataset of 115 problems represents a good size for an initial benchmark in this domain.
>
> ### Comment 2: Evaluation metric is based on compilation and verification success, not test case passing. Due to the nature of the problems (which are presented without completed solutions in the original data source), the authors resort to compilation and JPF based checks to measure success. However, the best verification for code is running against a test suite as code that can compile and pass JPF checks could still be wrong.
> ### Q2: Why is compilation success + JPF verification a sufficient measure of performance (instead of say compilation + JPF + running code against test suite)?
>
> ### Response:
> Extending the oracle with dynamic testing mechanisms, such as runtime schedule exploration or stress testing, would potentially enable more comprehensive correctness evaluation. However, dynamic testing of concurrent programs is highly nondeterministic and resource-intensive, especially across 115 benchmark programs, and designing a scalable and fair dynamic oracle requires careful engineering.
>
> In this work, we rely on model checking (via JPF), which provides a formal and systematic method for verifying concurrent programs by exploring thread interleavings, detecting data races, and identifying deadlocks. Compared to dynamic test suites, model checking offers a much stronger and more exhaustive guarantee of correctness for multithreaded code.
>
> Adding properties to check (in the form of assertions, which JPF can check) would make the evaluation even stronger. In future work we plan to investigate using LLMs to generate such properties (a research area in itself).

---

> > ### Author Response · Authors · 2025-11-20
> > **Response to the Question 3 of Reviewer oHT5**
> >
> > ### Q3: Is the proposed test set (just 43 problems) a statistically significant sample size for estimating concurrent code generation capabilities?
> >
> > Response:
> > We appreciate the reviewer’s question. Although the benchmark starts with 43 base programs, we additionally constructed 72 mutated variants, resulting in 115 total programs that collectively span 13 representative categories of concurrency problems. These categories were deliberately selected to capture the breadth of widely used synchronization primitives and multithreading patterns. To contextualize the scale, widely used code-generation benchmarks such as HumanEval include around 160 problems, showing that benchmarks of this magnitude are common and effective in practice. Given the substantially higher complexity of multithreaded programming, a dataset of 115 problems represents a good size for an initial benchmark in this domain.

---

> > > ### Author Response · Authors · 2025-11-21
> > > **Result of the updated experiments**
> > >
> > > The table reports Pass@1 accuracy for four prompt categories: (1) Original 43 Problems, which represent the full set included in prior evaluation; (2) 24 Mutation-Eligible Problems, the subset selected for prompt mutation; (3) 72 Mutated Problems, generated from that subset; and (4) All 115 Problems, combining original and mutated versions. This structure allows comparison across the different prompt groups and highlights how models behave under each configuration.
> > >
> > > Across the 22 LLMs evaluated, the results show a balanced outcome. Some models—such as gpt-5, claude-opus-4-1, and gpt-4o—obtain higher pass rates on the mutated prompts, whereas others—such as qwen3-32b, wizardcoder-33b, and phi-4-14b—perform better on the original prompts. This distribution suggests that the mutated prompts preserve the overall difficulty profile while providing additional variation. The combined Pass@1 results over all 115 prompts also follow consistent trends across models.
> > >
> > >
> > > ## LLM Pass@1 Rate Comparison
> > >
> > > | LLM | Original 43 Problems | 24 Mutation-Eligible Problems | 72 Mutated Problems | All 115 Problems |
> > > | :--- | :--- | :--- | :--- | :--- |
> > > | gpt-5 | 72.09% (31/43) | 83.33% (20/24) | 79.17% (57/72) | 76.52% (88/115) |
> > > | claude-opus-4-1 | 62.79% (27/43) | 58.33% (14/24) | 70.83% (51/72) | 67.83% (78/115) |
> > > | gpt-4o | 53.49% (23/43) | 50.00% (12/24) | 68.06% (49/72) | 62.61% (72/115) |
> > > | llama3.3:70b | 46.51% (20/43) | 45.83% (11/24) | 48.61% (35/72) | 47.83% (55/115) |
> > > | qwen3:32b | 41.86% (18/43) | 62.50% (15/24) | 41.67% (30/72) | 41.74% (48/115) |
> > > | codestral:22b | 37.21% (16/43) | 29.17% (7/24) | 44.44% (32/72) | 41.74% (48/115) |
> > > | wizardcoder:33b | 34.88% (15/43) | 41.67% (10/24) | 37.50% (27/72) | 36.52% (42/115) |
> > > | phi4:14b | 44.19% (19/43) | 37.50% (9/24) | 27.78% (20/72) | 33.91% (39/115) |
> > > | deepseek-r1:32b | 27.91% (12/43) | 29.17% (7/24) | 37.50% (27/72) | 33.91% (39/115) |
> > > | phind-codellama:34b | 25.58% (11/43) | 41.67% (10/24) | 37.50% (27/72) | 33.04% (38/115) |
> > > | gemma2:27b | 27.91% (12/43) | 16.67% (4/24) | 34.72% (25/72) | 32.17% (37/115) |
> > > | opencoder:8b | 34.88% (15/43) | 33.33% (8/24) | 29.17% (21/72) | 31.30% (36/115) |
> > > | mixtral:8x7b | 30.23% (13/43) | 25.00% (6/24) | 26.39% (19/72) | 27.83% (32/115) |
> > > | codeqwen:7b | 13.95% (6/43) | 12.50% (3/24) | 26.39% (19/72) | 21.74% (25/115) |
> > > | dolphin3:8b | 23.26% (10/43) | 20.83% (5/24) | 18.06% (13/72) | 20.00% (23/115) |
> > > | magicoder:7b | 16.28% (7/43) | 4.17% (1/24) | 11.11% (8/72) | 13.04% (15/115) |
> > > | vicuna:33b | 11.63% (5/43) | 8.33% (2/24) | 5.56% (4/72) | 7.83% (9/115) |
> > > | llava:34b | 9.30% (4/43) | 16.67% (4/24) | 4.17% (3/72) | 6.09% (7/115) |
> > > | starcoder2:15b | 6.98% (3/43) | 12.50% (3/24) | 0.00% (0/72) | 2.61% (3/115) |
> > > | zephyr:7b | 4.65% (2/43) | 4.17% (1/24) | 2.78% (2/72) | 3.48% (4/115) |
> > > | mistral:7b | 4.65% (2/43) | 8.33% (2/24) | 2.78% (2/72) | 3.48% (4/115) |
> > > | codellama:34b | 0.00% (0/43) | 0.00% (0/24) | 2.78% (2/72) | 1.74% (2/115) |
> > >
> > > ---

---

> > > > ### Comment · Reviewer_oHT5 · 2025-11-27
> > > >
> > > > I would like to thank the authors for their tremendous effort in answering our questions and extending the benchmark. While the benchmark is certainly interesting, I believe the questions of contamination and the nature of evaluation curation (how many annotators, inter-annotator agreement, etc.) are critically missing. I will maintain my score.

---

> > > > > ### Author Response · Authors · 2025-11-27
> > > > > **Response to the comment on contamination risk and the nature of evaluation curation**
> > > > >
> > > > > Thank you for raising this concern. We have now added a clear discussion of contamination risk: the textbook contains only fragmented, non-runnable snippets, so all 43 base problems required substantial reconstruction, refactoring, and specification work by the authors. Two authors independently designed the problem statements and ground-truth implementations, and when disagreements arose, they were resolved through discussion with the other authors. The 72 mutated variants further introduce structural and semantic variations do not present in the textbook, substantially reducing the likelihood that any model could rely on memorization. We hope this addresses the original concern, and we would appreciate clarification if there is a specific remaining aspect of contamination that you believe is still unresolved.
> > > > >
> > > > > Regarding the nature of the evaluation curation, two authors created and validated all ground-truth solutions and prompt specifications, and any disagreements were resolved through discussion with the other authors. We will make this process explicit in the paper. If there are additional details the reviewer believes would strengthen the clarity of our curation procedure, we would be grateful for the guidance, as this point was not raised in the initial review and we want to ensure we are addressing all concerns as thoroughly as possible.

---

### Official Review · Reviewer_bG35 · 2025-11-01

**Soundness:** 2
**Presentation:** 3
**Contribution:** 3
**Rating:** 2
**Confidence:** 5

**Summary:**

This work introduces CONCUR, a benchmark specifically designed to evaluate large language models on concurrent code generation. It contains 43 carefully curated multi-threading problems with structured prompts and validated Java implementations. Unlike existing benchmarks focusing on sequential code, CONCUR leverages Java PathFinder (JPF) for dynamic, exhaustive verification of generated programs, detecting concurrency errors such as deadlocks, race conditions, and single-thread violations. By integrating structured prompts, ground-truth solutions, and dynamic model checking, CONCUR provides a rigorous and reproducible framework to systematically assess LLMs’ ability to produce correct multi-threaded programs.

**Strengths:**

1. Novelty in Benchmark Design: The paper introduces CONCUR, the first benchmark specifically targeting multi-threaded code generation, filling a gap left by prior benchmarks that focus only on sequential programs.

2. Concurrency-Aware Problem Construction: The benchmark is carefully designed to enforce multi-threading features and concurrency-specific requirements, ensuring that generated programs must exhibit correct thread behavior and handle potential concurrency issues.

3. Clear Presentation and Comprehensive Validation: The paper is well-structured and clearly written, and it demonstrates the benchmark’s effectiveness by evaluating LLMs of various sizes and architectures, providing strong evidence for its reliability in systematically assessing concurrent code generation.

**Weaknesses:**

1. The benchmark includes only 43 Java programs, which is a relatively small number and may limit its coverage of diverse concurrent programming scenarios.

2. Although prompts for each program are provided in the public repository, the programs themselves are simple in functionality and description, which may not effectively evaluate LLMs’ ability to generate complex multi-threaded code.

**Questions:**

1. Could the benchmark be expanded with additional test programs to enhance coverage and diversity of concurrent scenarios?

2. Could the authors discuss the complexity of program functionality and explain the code selection process in more detail? Additionally, could they consider including more complex, real-world concurrent programs that better reflect practical programming scenarios?

3. Could the benchmark extend the oracle to include dynamic testing oracles, if feasible, for more comprehensive correctness evaluation?

---

> ### Author Response · Authors · 2025-11-20
> **Response to the comments of Reviewer bG35**
>
> ### Comment 1: The benchmark includes only 43 Java programs, which is a relatively small number and may limit its coverage of diverse concurrent programming scenarios.
> ### Q1: Could the benchmark be expanded with additional test programs to enhance coverage and diversity of concurrent scenarios?
>
> ### Response:
> We appreciate the reviewer’s concern regarding the number of benchmark programs. While the core benchmark consists of 43 concurrency problems, we have additionally constructed 72 mutated variants derived from these base problems to increase the size of the benchmark. These mutations were built by leveraging Gemini prompted to generate variations of the problem descriptions, which we validated manually, resulting in a benchmark of 115 total problems. We generated three mutants for each original problem, and after manual validation we ended up with meaningful mutants for 24 problems without changing their original semantics.
>
> Also, note that the problem set provides broad coverage of known multithreaded programming patterns and pitfalls. Across the original and mutated problems, the benchmark captures the following representative categories of concurrency constructs and issues:
>
> - synchronized
> - volatile
> - Lock interface
> - ReentrantLock
> - lock() / unlock()
> - tryLock()
> - Semaphore
> - CountDownLatch
> - Atomic* classes
> - BlockingQueue
> - Thread
> - Runnable / Callable
> - ExecutorService
>
> Our expanded benchmark allows us to systematically test LLM behavior across a diverse and realistic spectrum of multithreaded scenarios. Thus, although the number of original source programs is 43, the final benchmark includes 115 problems, offering a comprehensive evaluation of concurrent programming capabilities.
>
> To contextualize the scale, widely used code-generation benchmarks such as HumanEval include around 160 problems, showing that benchmarks of this magnitude are common and effective in practice. Given the substantially higher complexity of multithreaded programming,we believe a dataset of 115 problems represents a good size for an initial benchmark in this domain.
>
>
>
> ### Comment 2: Although prompts for each program are provided in the public repository, the programs themselves are simple in functionality and description, which may not effectively evaluate LLMs’ ability to generate complex multi-threaded code.
> ### Q2: Could the authors discuss the complexity of program functionality and explain the code selection process in more detail? Additionally, could they consider including more complex, real-world concurrent programs that better reflect practical programming scenarios?
>
> ### Response:
> While the prompts and corresponding programs in the benchmark are intentionally concise in their problem descriptions, the underlying ground-truth implementations are not simple. As noted above, the benchmark covers 13 representative categories of multithreaded constructs and concurrency issues, ensuring a broad and realistic challenge space.
>
> Our program selection and preparation process was designed to ensure both correctness and meaningful concurrency complexity. First, we collected source code examples from reputable reference books. Because these examples were often incomplete, we supplemented them as needed, for example, adding missing variable declarations, while preserving their intended multithreaded behavior. Second, we adjusted the maximum number of threads so that each program satisfied the minimum concurrency requirements implied by its description. Finally, we validated every program using Java PathFinder (JPF) to ensure that the supplemented and adjusted code contained no hidden concurrency or logical errors.
>
> For these categories, the ground-truth solutions range from 35 to 140 lines of code and include 2 to 29 functions, depending on the complexity of the concurrency mechanism being exercised. These implementations involve non-trivial synchronization logic, interactions between multiple threads, and correct usage of various concurrency primitives such as ReentrantLock, Semaphore, CountDownLatch, and ExecutorService.
>
> Our goal in this initial release was to ensure high-quality verified problems, accompanied by ground-truth implementations and model-checking results, which is non-trivial to construct. We prioritized quality over quantity in this first version.
>
> Also, we intentionally keep the prompts concise because, while they provide the necessary context, they do not explicitly describe how certain concurrency mechanisms—such as auxiliary locks—should be used. This design allows us to evaluate whether a model can infer the correct synchronization strategy rather than simply follow explicit instructions. A model that can correctly introduce and apply an auxiliary lock to avoid deadlocks demonstrates a stronger understanding of concurrency semantics. As a result, the benchmark provides a meaningful and rigorous evaluation of LLMs’ ability to generate realistic, multi-threaded Java programs.

---

> > ### Author Response · Authors · 2025-11-20
> > **Response to the Question 3 of Reviewer bG35**
> >
> > ### Q3: Could the benchmark extend the oracle to include dynamic testing oracles, if feasible, for more comprehensive correctness evaluation?
> >
> > ### Response:
> > Extending the oracle with dynamic testing mechanisms, such as runtime schedule exploration or stress testing, would typically enable more comprehensive correctness evaluation. However, dynamic testing of concurrent programs is highly nondeterministic and resource-intensive, especially across 115 benchmark programs, and designing a scalable and fair dynamic oracle requires careful engineering.
> >
> > In this work, we rely on model checking (via JPF), which provides a formal and systematic method for verifying concurrency programs by exploring *all the possible* thread interleavings, detecting data races, and identifying deadlocks. Compared to dynamic test suites, model checking offers a much stronger and more exhaustive guarantee of correctness for multithreaded code.

---

> > > ### Author Response · Authors · 2025-11-21
> > > **Result of the updated experiments**
> > >
> > > The table reports Pass@1 accuracy for four prompt categories: (1) Original 43 Problems, which represent the full set included in prior evaluation; (2) 24 Mutation-Eligible Problems, the subset selected for prompt mutation; (3) 72 Mutated Problems, generated from that subset; and (4) All 115 Problems, combining original and mutated versions. This structure allows comparison across the different prompt groups and highlights how models behave under each configuration.
> > >
> > > Across the 22 LLMs evaluated, the results show a balanced outcome. Some models—such as gpt-5, claude-opus-4-1, and gpt-4o—obtain higher pass rates on the mutated prompts, whereas others—such as qwen3-32b, wizardcoder-33b, and phi-4-14b—perform better on the original prompts. This distribution suggests that the mutated prompts preserve the overall difficulty profile while providing additional variation. The combined Pass@1 results over all 115 prompts also follow consistent trends across models.
> > >
> > >
> > > ## LLM Pass@1 Rate Comparison
> > >
> > > | LLM | Original 43 Problems | 24 Mutation-Eligible Problems | 72 Mutated Problems | All 115 Problems |
> > > | :--- | :--- | :--- | :--- | :--- |
> > > | gpt-5 | 72.09% (31/43) | 83.33% (20/24) | 79.17% (57/72) | 76.52% (88/115) |
> > > | claude-opus-4-1 | 62.79% (27/43) | 58.33% (14/24) | 70.83% (51/72) | 67.83% (78/115) |
> > > | gpt-4o | 53.49% (23/43) | 50.00% (12/24) | 68.06% (49/72) | 62.61% (72/115) |
> > > | llama3.3:70b | 46.51% (20/43) | 45.83% (11/24) | 48.61% (35/72) | 47.83% (55/115) |
> > > | qwen3:32b | 41.86% (18/43) | 62.50% (15/24) | 41.67% (30/72) | 41.74% (48/115) |
> > > | codestral:22b | 37.21% (16/43) | 29.17% (7/24) | 44.44% (32/72) | 41.74% (48/115) |
> > > | wizardcoder:33b | 34.88% (15/43) | 41.67% (10/24) | 37.50% (27/72) | 36.52% (42/115) |
> > > | phi4:14b | 44.19% (19/43) | 37.50% (9/24) | 27.78% (20/72) | 33.91% (39/115) |
> > > | deepseek-r1:32b | 27.91% (12/43) | 29.17% (7/24) | 37.50% (27/72) | 33.91% (39/115) |
> > > | phind-codellama:34b | 25.58% (11/43) | 41.67% (10/24) | 37.50% (27/72) | 33.04% (38/115) |
> > > | gemma2:27b | 27.91% (12/43) | 16.67% (4/24) | 34.72% (25/72) | 32.17% (37/115) |
> > > | opencoder:8b | 34.88% (15/43) | 33.33% (8/24) | 29.17% (21/72) | 31.30% (36/115) |
> > > | mixtral:8x7b | 30.23% (13/43) | 25.00% (6/24) | 26.39% (19/72) | 27.83% (32/115) |
> > > | codeqwen:7b | 13.95% (6/43) | 12.50% (3/24) | 26.39% (19/72) | 21.74% (25/115) |
> > > | dolphin3:8b | 23.26% (10/43) | 20.83% (5/24) | 18.06% (13/72) | 20.00% (23/115) |
> > > | magicoder:7b | 16.28% (7/43) | 4.17% (1/24) | 11.11% (8/72) | 13.04% (15/115) |
> > > | vicuna:33b | 11.63% (5/43) | 8.33% (2/24) | 5.56% (4/72) | 7.83% (9/115) |
> > > | llava:34b | 9.30% (4/43) | 16.67% (4/24) | 4.17% (3/72) | 6.09% (7/115) |
> > > | starcoder2:15b | 6.98% (3/43) | 12.50% (3/24) | 0.00% (0/72) | 2.61% (3/115) |
> > > | zephyr:7b | 4.65% (2/43) | 4.17% (1/24) | 2.78% (2/72) | 3.48% (4/115) |
> > > | mistral:7b | 4.65% (2/43) | 8.33% (2/24) | 2.78% (2/72) | 3.48% (4/115) |
> > > | codellama:34b | 0.00% (0/43) | 0.00% (0/24) | 2.78% (2/72) | 1.74% (2/115) |
> > >
> > > ---

---

### Meta-Review · Area_Chair_ZaCX · 2025-12-31

**Summary:**

The paper introduces CONCUR, the first benchmark specifically targeting multi-threaded code generation, filling a gap left by prior benchmarks that focus only on sequential programs.

Reviewers (bG35, oHT5, DzG4, fAkt) liked how there are sufficient number of models evaluated (22), and that model checking is used to catch concurrency bugs rather than just relying on testing.

However, reviewers (bG35, oHT5, fAkt) are unsure if Java problems from a single text book is sufficient to capture the whole domain of concurrency programming. The initial number of problems was also small in the 40s, and later expanded through mutation, which the reviewers did not further evaluate. Their construction method of using textbook problems also raises problems of contamination (oHT5), although the relative age of the text and language (Java) may offer some protection as well. Another weakness that is still unclear is if this evaluation method actually catches functionally incorrect code that nevertheless don't have issues checked for JPF (bG35, fAkt).

**Reviewer Concerns:**

Unfortunately most reviewers (bG35, oHT5, fAkt) did not further review the increase in size and mutation improvements, but this is unlikely to affect their decisions since their main points still stand, whereas mutation to increase data size is not really fully increasing the data size.

**Reviewer Scores:**

2226

---

### Decision · Program_Chairs · 2026-01-26

Reject